# LEARNING TO DEFENSE BY LEARNING TO ATTACK

## ABSTRACT

Adversarial training provides a principled approach for training robust neural networks. From an optimization perspective, the adversarial training is essentially solving a minimax robust optimization problem. The outer minimization is trying to learn a robust classifier, while the inner maximization is trying to generate adversarial samples. Unfortunately, such a minimax problem is very difficult to solve due to the lack of convex-concave structure. This work proposes a new adversarial training method based on a generic learning-to-learn (L2L) framework. Specifically, instead of applying the existing hand-designed algorithms for the inner problem, we learn an optimizer, which is parametrized as a convolutional neural network. At the same time, a robust classifier is learned to defense the adversarial attack generated by the learned optimizer. Our experiments over CIFAR-10 and CIFAR-100 datasets demonstrate that the L2L outperforms existing adversarial training methods in both classification accuracy and computational efficiency. Moreover, our L2L framework can be extended to the generative adversarial imitation learning and stabilize the training.

## 1 INTRODUCTION

This decade has witnessed great breakthroughs in deep learning in a variety of applications, such as computer vision (Taigman et al., 2014; Girshick et al., 2014; He et al., 2016; Liu et al., 2017). Recent studies (Szegedy et al., 2013), however, show that most of these deep learning models are very vulnerable to adversarial attacks. Specifically, by injecting a small perturbation to a normal sample, one can obtain an adversarial example. Although the adversarial example is semantically indistinguishable from the normal one, it can fool deep learning models and undermine the security of deep learning, causing reliability problems in autonomous driving, biometric authentication, etc.

Researchers have devoted many efforts to studying efficient adversarial attack and defense (Szegedy et al., 2013; Goodfellow et al., 2014b; Nguyen et al., 2015; Zheng et al., 2016; Madry et al., 2017; Carlini and Wagner, 2017). There is a growing body of work on generating adversarial examples, e.g., fast gradient sign method (FGSM, Goodfellow et al. (2014b)), projected gradient method (PGM, Kurakin et al. (2016)), Carlini-Wagner (CW, Paszke et al. (2017)) etc. As for defense, Goodfellow et al. (2014b) propose to robustify the network by adversarial training, which trains over the adversarial examples and still requires the network to output the correct label. Further, Madry et al. (2017) formalize the adversarial training as the following minimax optimization problem:

$$\min_{\boldsymbol{\theta}} \quad \frac{1}{n}\sum_{i=1}^{n}\big[\max_{\boldsymbol{\delta}_i \in \mathcal{B}} \ell(f(\boldsymbol{x}_i + \boldsymbol{\delta}_i; \boldsymbol{\theta}), y_i)\big], \tag{1}$$

where $\{(\boldsymbol{x}_i, y_i)\}_{i=1}^{n} \subset \mathbb{R}^d \times \mathcal{Y}$ are $n$ pairs of input feature and the corresponding label, $\ell$ denotes a loss function, $f(\cdot; \boldsymbol{\theta})$ denotes the neural network with parameter $\boldsymbol{\theta}$, and $\boldsymbol{\delta}_i \in \mathcal{B}$ denotes the perturbation for $\boldsymbol{x}_i$ in constraint $\mathcal{B}$. The existing literature on the optimization also refers to $\boldsymbol{\theta}$ as the primal variable and $\boldsymbol{\delta}_i$'s as the dual variables. Different from the well-studied convex-concave problem[1], problem (1) is very challenging since $\ell$ is nonconvex in $\boldsymbol{\theta}$ and nonconcave in $\boldsymbol{\delta}$. As a result, there may be many equilibria, and majority of them are unstable. In the existing optimization literature, there is no algorithm to converge to a stable equilibrium with theoretical guarantees. Empirically, the existing primal-dual algorithms may perform poorly for solving (1).

Minimax formulation (1) naturally provides us with a unified perspective on prior works of adversarial training. Such a minimax problem consists of two optimization problems, an inner maximization problem and an outer minimization problem: The inner problem targets on finding an optimal attack

---

[1]Loss function $\ell(\boldsymbol{\theta}; \boldsymbol{\delta})$ is convex in primal variable $\boldsymbol{\theta}$ and concave in dual variable $\boldsymbol{\delta}$.

for a given data point $(\boldsymbol{x}, y)$ maximizing the loss, which essentially is the adversarial attack; The outer problem aims to find a $\boldsymbol{\theta}$ so that the loss given by the inner problem is minimized.

For solving (1), Goodfellow et al. (2014b) propose to use FGSM to solve the inner problem. Kurakin et al. (2016) then find that FGSM with true label predicted suffers from a "label leaking" effect, which can ruin the adversarial training. Madry et al. (2017) further suggest to solve the inner problem by PGM and obtain a better result than FGSM, since FGSM essentially is one iteration PGM. However, adversarial training needs to find a $\boldsymbol{\delta}_i$ for each $(\boldsymbol{x}_i, y_i)$, thus the dimension of the overall search space for all data is substantial, which makes the computation expensive.

Instead, we propose a new learning-to-learn (L2L) framework that provides a more principled and *efficient* way for adversarial training. Specifically, we parameterize the optimizer of the inner maximization problem by a neural network denoted by $g(\mathcal{A}(\boldsymbol{x}, y, \boldsymbol{\theta}); \boldsymbol{\phi})$, where $\mathcal{A}(\boldsymbol{x}, y, \boldsymbol{\theta})$ denotes the input of the optimizer $g$. We also call the optimizer as the attacker network. Since the neural network is very powerful in function approximation, our parameterization ensures that $g$ is able to yield strong adversarial perturbations. Under our framework, instead of directly solving $\boldsymbol{\delta}_i$, we update the parameter $\boldsymbol{\phi}$ of $g$. Our training procedure becomes updating the parameters of two neural networks, which is very similar to generative adversarial network (GAN, Goodfellow et al. (2014a)). The proposed L2L is a generic framework and can be extended to other minimax optimization problems, e.g., generative adversarial imitation learning, which is studied in Section 4.

Different from the hand-designed methods that compute the adversarial perturbation for each individual sample using gradients from backpropagation, our methods generate the perturbations for all samples through the shared attacker $g$. This enables the attacker $g$ to learn potential common structures of the perturbations. Therefore, our method is capable of yielding strong perturbations and accelerating the training process. Furthermore, L2L framework is very *flexible*: we can either choose different input $\mathcal{A}(\boldsymbol{x}, y, \boldsymbol{\theta})$, or use different attacker architecture. For example, we can include gradient information in $\mathcal{A}(\boldsymbol{x}, y, \boldsymbol{\theta})$ and use a recurrent neural network (RNN) to mimic multi-step gradient-type methods. Instead of simply computing the high order information with finite difference approximation or multiple gradients, by parameterizing the algorithm as a neural network, our proposed methods can capture this information in a much smarter way (Finn et al., 2017). Our experiments demonstrate that our proposed methods not only *outperform* existing adversarial training methods, e.g., PGM training, but also enjoy the computational efficiency over CIFAR-10 and CIFAR-100 datasets (Krizhevsky and Hinton, 2009).

The research on L2L has a long history (Schmidhuber, 1987; 1992; 1993; Younger et al., 2001; Hochreiter et al., 2001; Andrychowicz et al., 2016). The basic idea is that one first models the updating formula of complicated optimization algorithms in a parametric form, and then uses some simple algorithms, e.g., stochastic gradient algorithm to learn the parameters of the optimizer. Among existing works, Hochreiter et al. (2001) propose a system allowing the output of backpropagation from one network to feed into an additional learning network, with both networks trained jointly; Based on this, Andrychowicz et al. (2016) further show that the design of an optimization algorithm can be cast as a learning problem. Specifically, they use long short-term memory RNNs to model the algorithm and allow the RNNs to exploit structure in the problems of interest in an automatic way, which is undoubtedly one of the most popular methods for learning-to-learn.

However, there are two major drawbacks of the existing L2L methods: **(1)** It requires a large amount of datasets (or a large number of tasks in multi-task learning) to guarantee the learned optimizer to generalize, which significantly limits their applicability (most of the related works only consider the image encoding as the motivating application); **(2)** The number of layers/iterations in RNNs for modeling algorithms cannot be large to avoid significant computational burden in backpropagation.

Our contribution is that we fill the blank of the L2L framework in solving the minimax problem, and our proposed methods do not suffer from the aforementioned drawbacks: **(1)** The attacker $g$ with a different $\boldsymbol{\phi}$ essentially generates a different task/dataset. Therefore, for adversarial training, we have sufficiently many tasks for learning-to-learn; **(2)** The inner problem does not need a large scale RNN, and we use a convolutional neural network (CNN) or a length-two RNN (the sequence of length equals 2) as our attacker network, which eases the computation.

Our work is also related to GAN and dual-embedding (Dai et al., 2016). All three solve minimax problems and share some common ground. We discuss these works in Section 5.

**Notations**. Given $a \in \mathbb{R}$, denote $(a)_+$ as $\max(a, 0)$. Given $\boldsymbol{x}, \boldsymbol{y} \in \mathbb{R}^d$, denote $x_i$ as the $i$-th element of $\boldsymbol{x}$, $||\boldsymbol{x}||_\infty = \max_i |x_i|$ as $\ell_\infty$-norm of $\boldsymbol{x}$, and $\boldsymbol{x} \circ \boldsymbol{y} = [x_1 y_1, \cdots, x_d y_d]^\top$ as element-wise product.

## 2 METHOD

This paper focuses on $\ell_\infty$-norm attack. We define the $\ell_\infty$-ball with center $\mathbf{0}$ and radius $\epsilon$ by $\mathcal{B}(\epsilon) = \{\boldsymbol{\delta} \in \mathbb{R}^d : ||\boldsymbol{\delta}||_\infty \leq \epsilon\}$ and the corresponding projection as follows:

$$\Pi_{\mathcal{B}(\epsilon)}(\boldsymbol{\delta}) = \mathrm{sign}(\boldsymbol{\delta}) \circ \max(|\boldsymbol{\delta}|, \epsilon),$$

where $\mathrm{sign}$ and $\max$ are element-wise operators.

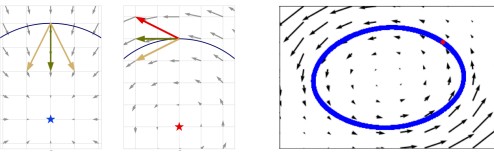

**Ideal  Reality  Limiting Cycle**
Figure 1: *Illustration for the hardness of problem* (1). *A wrong update direction leads to a limiting cycle and algorithms fail to converge. See Appendix D for more details.*

### 2.1 ADVERSARIAL TRAINING

The goal of adversarial training is to robustify neural networks. Recall that from a robust optimization perspective, given $n$ samples $\{(\boldsymbol{x}_i, y_i)\}_{i=1}^n$, where $\boldsymbol{x}_i$ is the $i$-th feature vector and $y_i$ is the corresponding label, adversarial training is reformulated as the following optimization problem (1):

$$\min_{\boldsymbol{\theta}} \quad \frac{1}{n} \sum_{i=1}^n \big[ \max_{\boldsymbol{\delta}_i \in \mathcal{B}} \ell(f(\boldsymbol{x}_i + \boldsymbol{\delta}_i; \boldsymbol{\theta}), y_i) \big],$$

where $f$ denotes the network with parameter $\boldsymbol{\theta}$, $\ell$ denotes a loss function, and $\epsilon$ is the maximum perturbation magnitude.

We first demonstrate the hardness of solving problem (1). Ideally, we want to obtain the optimal solution for the inner problem

$$\boldsymbol{\delta}_i^* := \mathrm{argmax}_{\boldsymbol{\delta}_i \in \mathcal{B}} \ell(f(\boldsymbol{x}_i + \boldsymbol{\delta}_i; \boldsymbol{\theta}), y_i).$$

However, note that the loss function $\ell(f(\boldsymbol{x}_i + \boldsymbol{\delta}_i; \boldsymbol{\theta}), y_i)$ is highly nonconcave in $\boldsymbol{\delta}_i$. Therefore, in reality the $\boldsymbol{\delta}_i$ we obtained is very unlikely to be the optimum $\boldsymbol{\delta}_i^*$. This then often leads to a highly unreliable or even completely wrong search direction, e.g.,

$$\langle \nabla_{\boldsymbol{\theta}} \ell(f(\boldsymbol{x}_i + \boldsymbol{\delta}_i; \boldsymbol{\theta}), y_i), \nabla_{\boldsymbol{\theta}} \ell(f(\boldsymbol{x}_i + \boldsymbol{\delta}_i^*; \boldsymbol{\theta}), y_i) \rangle < 0,$$

which may further results in a limiting cycle shown in Figure 1 (Detailed discussion is provided in Appendix D). This becomes even worse when sample noises exist.

In the existing literature, the standard pipeline of adversarial training is shown in Algorithm 1. Since the step of generating adversarial perturbation $\boldsymbol{\delta}_i$ in Algorithm 1 is intractable, most adversarial training methods adopt hand-designed algorithms. For example, Kurakin et al. (2016) propose to solve the inner problem approximately by first order methods such as PGM. Specifically, PGM iteratively updates the adversarial perturbation by the projected sign gradient ascent method for each sample: Given one sample $(\boldsymbol{x}_i, y_i)$, at the $t$-th iteration, PGM takes

---

**Algorithm 1** *Standard pipeline of adversarial training*

**Input:** $\{(\boldsymbol{x}_i, y_i)\}_{i=1}^n$: clean data, $\alpha$: learning rate, $N$: number of epochs, $\epsilon$: maximum perturbation magnitude.

**for** $t \leftarrow 1$ *to* $N$ **do**
    Sample a minibatch $\mathcal{M}_t$
    **for** $i$ *in* $\mathcal{M}_t$ **do**
        $\boldsymbol{\delta}_i \leftarrow \mathrm{argmax}_{\boldsymbol{\delta}_i \in \mathcal{B}(\epsilon)} \ell(f(\boldsymbol{x}_i + \boldsymbol{\delta}_i; \boldsymbol{\theta}), y_i)$
        Generate adversarial perturbation for $(\boldsymbol{x}_i, y_i)$
    $\boldsymbol{\theta} \leftarrow \boldsymbol{\theta} - \alpha \frac{1}{|\mathcal{M}_t|} \sum_{i \in \mathcal{M}_t} \nabla_{\boldsymbol{\theta}} \ell(f(\boldsymbol{x}_i + \boldsymbol{\delta}_i; \boldsymbol{\theta}), y_i)$
    Update $\boldsymbol{\theta}$ over adversarial data $\{(\boldsymbol{x}_i + \boldsymbol{\delta}_i, y_i)\}_{i \in \mathcal{M}_t}$

---

$$\boldsymbol{\delta}_i^t \leftarrow \Pi_{\mathcal{B}(\epsilon)}\big(\boldsymbol{\delta}_i^{t-1} + \eta \cdot \mathrm{sign}\big(\nabla_{\boldsymbol{x}} \ell(f(\widetilde{\boldsymbol{x}}_i^t; \boldsymbol{\theta}), y))\big)\big), \tag{2}$$

where $\widetilde{\boldsymbol{x}}_i^t = \boldsymbol{x}_i + \boldsymbol{\delta}_i^{t-1}$, $\eta$ is the perturbation step size, $T$ is a pre-defined total number of iterations, and $\boldsymbol{\delta}_i^0 = \mathbf{0}$, $t = 1, \cdots, T$. Finally PGM takes $\boldsymbol{\delta}_i = \boldsymbol{\delta}_i^T$. Note that FGSM essentially is one-iteration PGM. Besides, some works adopt other optimization methods, such as momentum gradient method (Dong et al., 2018), and L-BFGS (Tabacof and Valle, 2016). However, except for FGSM, all require numerous queries for gradients through backpropagation, which is computationally expensive.

### 2.2 LEARNING TO DEFENSE BY LEARNING TO ATTACK (L2L)

Since the objective function is nonconvex-nonconcave, there is no guarantee for hand-designed methods to perform well. Instead, we propose to learn an optimizer for the inner problem. Specifically, we parameterize the attacker by a Neural Network $g(\mathcal{A}(\boldsymbol{x}, y, \boldsymbol{\theta}); \boldsymbol{\phi})$, where $\mathcal{A}(\boldsymbol{x}, y, \boldsymbol{\theta})$, the

input of the network $g$, summaries the information of data and neural network $f(\cdot; \boldsymbol{\theta})$. We then convert problem (1) to

$$\min_{\boldsymbol{\theta}} \frac{1}{n} \sum_{i=1}^{n} \left[ \ell(f(\boldsymbol{x}_i + g(\mathcal{A}(\boldsymbol{x}_i, y_i, \boldsymbol{\theta}); \boldsymbol{\phi}^*); \boldsymbol{\theta}), y_i) \right], \tag{3}$$

where $\boldsymbol{\phi}^*$ is defined as the solution to the following optimization problem:

$$\boldsymbol{\phi}^* \in \underset{\boldsymbol{\phi}}{\operatorname{argmax}} \frac{1}{n} \sum_{i=1}^{n} \ell(f(\boldsymbol{x}_i + g(\mathcal{A}(\boldsymbol{x}_i, y_i, \boldsymbol{\theta}); \boldsymbol{\phi}); \boldsymbol{\theta}), y_i) \text{ subject to } g(\mathcal{A}(\boldsymbol{x}, y, \boldsymbol{\theta}); \boldsymbol{\phi}) \in \mathcal{B}(\epsilon).$$

Solving problem (3) naturally consists of two stages. In the first stage, the classifier $f$ aims to fit over all perturbed data; While in the second stage, given a certain $f$ obtained in the first stage, the attacker network $g$ targets on generating optimal perturbations under constraints $\boldsymbol{\delta}_i$'s $\in \mathcal{B}(\epsilon)$.

Since $\boldsymbol{\delta}_i = g(\mathcal{A}(\boldsymbol{x}_i, y_i; \boldsymbol{\theta}); \boldsymbol{\phi})$, constraints can be simply handled by a $\tanh$ activation function in the last layer of $g$. Specifically, because the magnitude of $\tanh$ output is bounded by 1, after we rescale the output by $\epsilon$, the output of $g$ automatically satisfies the constraints.

This framework is very flexible. We can choose different $\mathcal{A}(\boldsymbol{x}, y, \boldsymbol{\theta})$ as the input and mimic multi-step gradient algorithms shown in Figure 2. Here we provide three examples:

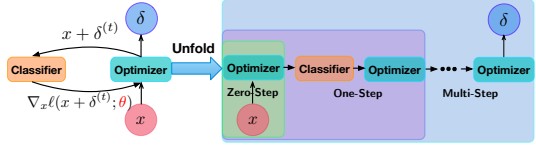

**Naive Attacker Network.** This is the simplest example among our methods, taking the original image $\boldsymbol{x}_i$ as the input, i.e.,

$$\mathcal{A}(\boldsymbol{x}_i, y_i, \boldsymbol{\theta}) = \boldsymbol{x}_i \quad \text{and} \quad \boldsymbol{\delta}_i = g(\boldsymbol{x}_i; \boldsymbol{\phi}).$$

Figure 2: *An illustration of L2L. We learn a neural network to model the algorithm for generating adversarial attack.*

Under this setting, L2L training is similar to GAN training. The major difference is that the generator in GAN yields synthetic data by transforming random noises, while the naive attacker network generates perturbations by transforming training samples.

**Gradient Attacker Network.** Motivated by hand-designed methods, e.g., FGSM, we design an attacker which takes the gradient information into computation. Specifically, we concatenate image $\boldsymbol{x}_i$ and gradient $\nabla_{\boldsymbol{x}} \ell(f(\boldsymbol{x}_i; \boldsymbol{\theta}), y_i)$ from backpropagation as the input of $g$, i.e.,

$$\mathcal{A}(\boldsymbol{x}_i, y_i, \boldsymbol{\theta}) = \left[ \boldsymbol{x}_i, \nabla_{\boldsymbol{x}} \ell(f(\boldsymbol{x}_i; \boldsymbol{\theta}), y_i) \right]$$

$$\text{and} \quad \boldsymbol{\delta}_i = g(\boldsymbol{x}_i, \nabla_{\boldsymbol{x}} \ell(f(\boldsymbol{x}_i; \boldsymbol{\theta}), y_i); \boldsymbol{\phi}).$$

Since more information is provided, we expect the attacker network to be more effective to learn and yield more powerful perturbations.

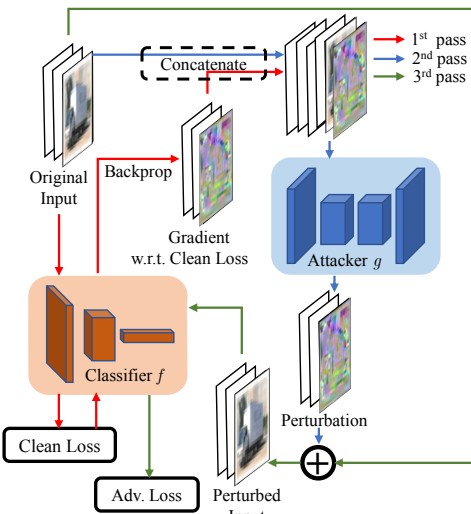

**Multi-Step Gradient Attacker Network.** We adapt the RNN to mimic a multi-step gradient update. Specifically, we use the gradient attacker network as the cell of RNN sharing the same parameter $\boldsymbol{\phi}$. As we mentioned earlier, the number of layers/iterations in the RNN for modeling algorithms cannot be very large so as to avoid significant computational burden in backpropagation. In this paper, we focus on a length-two RNN to mimic a two-step gradient update.

Figure 3: *The architecture of L2L adversarial training with gradient attacker network.*

The corresponding perturbation becomes:

$$\boldsymbol{\delta}_i = \Pi_{\mathcal{B}(\epsilon)} \left( \boldsymbol{\delta}_i^{(0)} + g(\widetilde{\boldsymbol{x}}_i, \nabla_{\boldsymbol{x}} \ell(f(\widetilde{\boldsymbol{x}}_i, y_i; \boldsymbol{\theta}); \boldsymbol{\phi})) \right),$$

where $\widetilde{\boldsymbol{x}}_i = \boldsymbol{x}_i + \boldsymbol{\delta}_i^{(0)}$ and $\boldsymbol{\delta}_i^{(0)} = g(\boldsymbol{x}_i, \nabla_{\boldsymbol{x}} \ell(f(\boldsymbol{x}_i, y_i; \boldsymbol{\theta}); \boldsymbol{\phi}))$.

Taking gradient attacker network as an example, Figure 3 illustrates how L2L works and jointly trains two networks: The first forward pass is used to obtain gradient of the classification loss over clean data; The second forward pass is used to generate perturbation $\boldsymbol{\delta}_i$ by attacker $g$; The third forward pass is used to calculate the adversarial loss $\ell$ in (3). Since our gradient attacker network only needs one backpropagation to query gradient, it amortizes the adversarial training cost, which leads to better computational efficiency. Moreover, L2L may adapt to the underlying optimization problem and yield better solution for the inner problem. The corresponding procedure of L2L is shown in Algorithm 2.

## 3 EXPERIMENTS

To demonstrate the effectiveness and computational efficiency of our methods, we compare our methods with PGM training over CIFAR-10 and CIFAR-100 datasets. All experiments are done in PyTorch with one NVIDIA 2080 Ti GPU.

For simplicity, we denote Plain Net as the classifier network trained over clean data only, PGM Net as the classifiers with PGM training, and Naive L2L, Grad L2L, and 2-Step L2L as classifiers using L2L training with corresponding attacker networks.

---

**Algorithm 2** *Learning-to-learn-based adversarial training with gradient attacker network*

---

**Input:** $\{(\boldsymbol{x}_i, y_i)\}_{i=1}^n$: clean data, $\alpha_1, \alpha_2$: learning rates, $N$: number of epochs, $\epsilon$: maximum perturbation magnitude.

**for** $t \leftarrow 1$ *to* $N$ **do**
    Sample a minibatch $\mathcal{M}_t$
    **for** $i$ *in* $\mathcal{M}_t$ **do**
        $\boldsymbol{u}_i \leftarrow \nabla_{\boldsymbol{x}} \ell(f(\boldsymbol{x}_i; \boldsymbol{\theta}), y_i)$
        $\boldsymbol{\delta}_i \leftarrow g(\boldsymbol{x}_i, \boldsymbol{u}_i; \boldsymbol{\phi})$
        Generate perturbation by $g$
    $\boldsymbol{\theta} \leftarrow \boldsymbol{\theta} - \alpha_1 \frac{1}{|\mathcal{M}_t|} \sum_{i \in \mathcal{M}_t} \nabla_{\boldsymbol{\theta}} \ell(f(\boldsymbol{x}_i + \boldsymbol{\delta}_i; \boldsymbol{\theta}), y_i)$
    Update $\boldsymbol{\theta}$ over adversarial data $\{(\boldsymbol{x}_i + \boldsymbol{\delta}_i, y_i)\}_{i \in \mathcal{M}_t}$
    $\boldsymbol{\phi} \leftarrow \boldsymbol{\phi} + \alpha_2 \frac{1}{|\mathcal{M}_t|} \sum_{i \in \mathcal{M}_t} \nabla_{\boldsymbol{\phi}} \ell(f(\boldsymbol{x}_i + \boldsymbol{\delta}_i \boldsymbol{\theta}), y_i)$
    Update $\boldsymbol{\phi}$ over adversarial data $\{(\boldsymbol{x}_i + \boldsymbol{\delta}_i, y_i)\}_{i \in \mathcal{M}_t}$

---

**Classifier Network.** All experiments adopt a 34-layer wide residual network (WRN-34-10, Zagoruyko and Komodakis (2016)) implemented by Zhang et al. (2019) as the classifier network. For each method, we train the classifier network from scratch. For Plain Nets, we adapt the same training procedure as Zagoruyko and Komodakis (2016). We use cross entropy as the loss function.

**Attacker Network.** Table 1 presents the architecture of our attacker network[2]. The ResBlock uses the same structure as the generator proposed in Miyato et al. (2018). The detailed structure of ResBlock is provided in Appendix B. Batch normalization (BN) and activations, e.g., ReLU and $\tanh$, are applied when specified. The $\tanh$ function easily makes the output of attacker satisfy the constraints.

Table 1: *Attacker network architecture: $k, c, s, p$ denote the kernel size, output channels, stride and padding parameters of convolutional layers, respectively.*

| | |
|---|---|
| Conv: | $[k = 3 \times 3, c = 64, s = 1, p = 1]$, BN+ReLU |
| ResBlock: | $[k = 3 \times 3, c = 128, s = 1, p = 1]$ |
| ResBlock: | $[k = 3 \times 3, c = 256, s = 1, p = 1]$ |
| ResBlock: | $[k = 3 \times 3, c = 128, s = 1, p = 1]$ |
| Conv: | $[k = 3 \times 3, c = 3, s = 1, p = 1]$, $\tanh$ |

**White-box and Black-box.** We compare different methods under both white-box and black-box settings. Under the white-box setting, attackers can access all parameters of target models and generate adversarial examples based on the models; whereas under the black-box setting, accessing parameters is prohibited. Therefore, we adopt the standard transfer attack method from Liu et al. (2016). Due to the space limit, we leave results of the black-box setting in Appendix A.

**Robust Evaluation.** We evaluate the robustness of the networks by PGM and CW attacks with the maximum perturbation magnitude $\epsilon = 0.031$ (after rescaling the pixels to $[0, 1]$) over CIFAR-10 and CIFAR-100. For PGM attack, we use 20 and 100-iteration PGM with a perturbation step size $\eta = 0.003$, and for each sample we initial the adversary randomly in the $\ell_\infty$ ball of radius $\epsilon$ centered at the natural sample. For CW attack, we adopt the implementation from Paszke et al. (2017), and set the maximum number of iterations as 100. For each method, we repeat 5 runs with different random initial seed and report the *worst result*. For CIFAR-10, we also evaluate the robustness of our Grad L2L and 2-Step L2L networks using random attacks, for which we uniformly sample 10000 perturbations in the $\ell_\infty$ ball with radius $\epsilon = 0.031$ adding to each test sample. We also evaluate the robustness of our Grad L2L and 2-Step L2L networks under their own attackers.

**L2L.** To update classifier's parameter $\boldsymbol{\theta}$, we use the SGD algorithm with Polyak's momentum (parameter 0.9, Liu et al. (2018)) and weight decay (parameter $2 \times 10^{-4}$, Krogh and Hertz (1992)); To update the attacker's parameter $\boldsymbol{\phi}$, we use the Adam optimizer (parameter $[0.9, 0.999]$, Kingma and Ba (2014)) and weight decay (parameter $2 \times 10^{-4}$) so that it adaptively balances the inner and outer optimization. In addition, we train the whole network for 100 epochs and set the initial learning rate for SGD as 0.1, the learning rate decay parameter as 0.1 with decay schedule [30,60,90], and the initial learning rate for Adam optimizer as 0.001 without learning rate decay.

---

[2]We provide another attacker architecture with down-sampling modules in the Appendix B. With such an attacker, L2L adversarial training is less stable, but much faster.

**PGM Adversarial Training.** For CIFAR-10, we directly report the result from Madry et al. (2017) as the baseline; For CIFAR-100, we train a PGM Net as a baseline. To update the parameter $\theta$, we use the same configuration of SGD algorithm in L2L. Moreover, we adopt the setting from Madry et al. (2017), that is, we first use 10-iteration PGM with the perturbation step size $0.007$ in (2) to generate the adversarial samples, and then train the PGM Net over these samples.

Table 2: Results of different defense methods under the white-box setting.

| Defense Method | Attack | Dataset | Clean Accuracy | Robust Accuracy |
|---|---|---|---|---|
| Zheng et al. (2016) | PGM-20 | CIFAR10 | 94.64% | 0.15% |
| Kurakin et al. (2016) | PGM-20 | CIFAR10 | 85.25% | 45.89% |
| Madry et al. (2017) | PGM-20 | CIFAR10 | 87.30% | 47.04% |
| Naive L2L | PGM-20 | CIFAR10 | 94.53% | 0.01% |
| Grad L2L | PGM-20 | CIFAR10 | 85.84% | 51.17% |
| 2-Step L2L | PGM-20 | CIFAR10 | 85.35% | 54.32% |
| Grad L2L | PGM-100 | CIFAR10 | 85.84% | 47.72% |
| 2-Step L2L | PGM-100 | CIFAR10 | 85.35% | 52.12% |
| Grad L2L | CW | CIFAR10 | 85.84% | 53.5% |
| 2-Step L2L | CW | CIFAR10 | 85.35% | 57.07% |
| Grad L2L | Random | CIFAR10 | 85.84% | 82.67% |
| 2-Step L2L | Random | CIFAR10 | 85.35% | 83.10% |
| Grad L2L | Grad L2L | CIFAR10 | 85.84% | 49.68% |
| 2-Step L2L | 2-Step L2L | CIFAR10 | 85.35% | 52.71% |
| PGM Net | PGM-20 | CIFAR100 | 62.68% | 23.75% |
| Grad L2L | PGM-20 | CIFAR100 | 62.18% | 28.67% |
| 2-Step L2L | PGM-20 | CIFAR100 | 60.95% | 31.03% |
| PGM Net | PGM-100 | CIFAR100 | 62.68% | 22.06% |
| Grad L2L | PGM-100 | CIFAR100 | 62.18% | 26.69% |
| 2-Step L2L | PGM-100 | CIFAR100 | 60.95% | 29.75% |
| PGM Net | CW | CIFAR100 | 62.68% | 25.95% |
| Grad L2L | CW | CIFAR100 | 62.18% | 29.65% |
| 2-Step L2L | CW | CIFAR100 | 60.95% | 32.28% |

Table 3: *Running time for one epoch over CIFAR-10 ($1^{st}$ Row) and CIFAR-100 ($2^{nd}$ Row). (Unit: s)*

| Plain Net | PGM Net | Naive L2L | Grad L2L | 2-Step L2L |
|---|---|---|---|---|
| $106.5 \pm 1.5$ | $1310.8 \pm 14.2$ | $293.7 \pm 3.1$ | $617.5 \pm 6.1$ | $805.1 \pm 8.1$ |
| $106.9 \pm 1.4$ | $1354.8 \pm 14.1$ | $310.0 \pm 2.9$ | $623.1 \pm 6.3$ | $824.7 \pm 8.4$ |

**Experiment Results.** Table 2 shows the results of all methods over CIFAR-10 and CIFAR-100 under the white-box setting. As can be seen, without gradient information, Naive L2L is vulnerable to the PGM attack. However, when the attacker utilizes the gradient information, Grad L2L and 2-Step L2L *significantly outperform* the PGM Net over CIFAR-10 and CIFAR-100, with a small loss for the clean accuracy. From the experiments on CIFAR-10, our Grad L2L and 2-Step L2L are robust to random attacks, where the accuracy is only slightly lower than the clean accuracy. Furthermore, the accuracy of our Grad/2-Step L2L model under the Grad/2-Step L2L attacker is comparable to the accuracy under PGM attacks, which shows

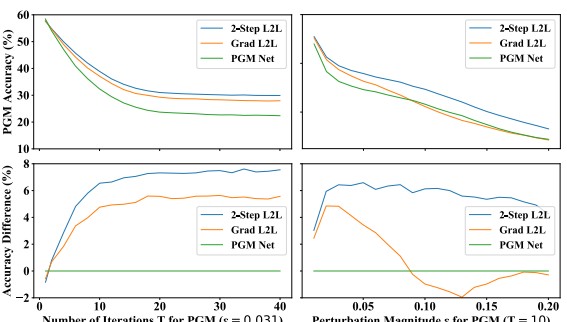

Figure 4: *Robust accuracy against perturbation magnitudes and number of iterations of PGM over CIFAR-100. Top two figures show the absolute accuracy over adversarial samples; the bottom two show the difference accuracy (PGM Net as baseline). For larger perturbation magnitude see Appendix E*

that L2L attackers are able to generate strong attacks. As can be seen, PGM-100 is stronger than Grad L2L attacker ($47.72\%$ vs. $49.68\%$), but similar to the 2-Step L2L attacker ($52.07\%$ vs. $52.71\%$). This means the strength of 2-Step L2L attacker is much stronger than Grad L2L attacker and explains why 2-Step L2L is stronger than Grad L2L and PGM net. In addition, Table 3 shows the one epoch running time of all methods over CIFAR-10 and CIFAR-100. As can be seen, Grad L2L and 2-Step L2L is much faster than PGM Net. By further comparing the accuracy of Grad/2-Step L2L and PGM Net in Table 2, we find that L2L enjoys computational efficiency. In addition,

Figure 4 presents the robust accuracy against number of iterations (fixed perturbation magnitude $\epsilon = 0.031$) and perturbation magnitude (fixed number of iterations $T = 10$). As can be seen, 2-Step L2L is much more robust than PGM Net.

**Visualization of Adversarial Examples.** Figure 5 provides an illustrative example of the adversarial perturbations generated by FGSM, PGM-20 and 2-Step L2L attacker for a *cat* over CIFAR-10. As can be seen, the attacks for these two networks are very different. Moreover, the perturbation generated by the 2-Step L2L attacker is much more smooth than FGSM and PGM. In this example, 2-Step L2L labels all adversarial samples correctly; whereas the PGM Net is fooled by the PGM-20 attack and misclassifies it as a *dog*.

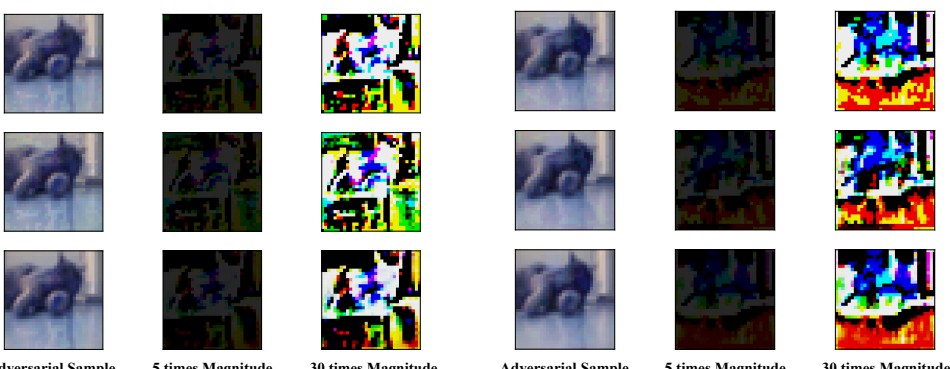



Adversarial Sample     5 times Magnitude     30 times Magnitude        Adversarial Sample     5 times Magnitude     30 times Magnitude

(a) Adversarial samples from 2-Step L2L.        (b) Adversarial samples from PGM Net.



Figure 5: *Illustrative examples of FGSM (Top), PGM-20 (Mid), and 2-Step L2L (Bottom) perturbations for a cat under PGM Net and 2-Step L2L with perturbation magnitude $\epsilon = 0.031$.*

# 4 EXTENSION: GENERATIVE ADVERSARIAL IMITATION LEARNING

Our proposed L2L is a *generic* framework and can be applied to a *broad* class of minimax optimization problems. Here, we investigate the generative adversarial imitation learning (GAIL, Ho and Ermon (2016)). GAIL aims to learn a policy from expert's behavior, by recovering the expert's cost function and extracting a policy from the recovered cost function, which can also be formulated as a minimax optimization problem. We then show that under the L2L framework, we stabilize the GAIL training by replacing the inner optimizer with a neural network.

**Original GAIL.** GAIL solves the following minimax optimization problem:

$$\min_{\theta_\pi} \max_{\theta_D} \mathbb{E}_{s,a\sim\pi(s;\theta_\pi)}[\log\left(D(s,a;\theta_D)\right)] + \mathbb{E}_{\widetilde{s},\widetilde{a}\sim\pi_E}[\log(1 - D(\widetilde{s},\widetilde{a};\theta_D))] - \lambda H(\pi), \quad (4)$$

where $\pi(\cdot;\theta_\pi)$ is the trained policy parameterized by $\theta_\pi$, $\pi_E$ denotes the expert policy, $D(\cdot,\cdot;\theta_D)$ is the discriminator parameterized by $\theta_D$, $\lambda H(\pi)$ denotes a entropy regularizer with tuning parameter $\lambda$, $(s,a)$ and $(\widetilde{s},\widetilde{a})$ denote the state-action for the trained policy and expert policy, respectively. By optimizing 4, the discriminator $D$ distinguishes the state-action $(s,a)$ generated from the learned policy $\pi$ with the sampled trajectories $(\widetilde{s},\widetilde{a})$ generated from some expert policy $\pi_E$. In the original GAIL training, for each iteration, we update the parameter of $D$, $\theta_D$, by stochastic gradient ascend and then update $\theta_\pi$ by the trust region policy optimization (TRPO, Schulman et al. (2015)).

**GAIL with L2L.** Similar to the adversarial training with L2L, we apply our L2L framework to GAIL by parameterizing the inner optimizer as a neural network $U(;\theta_U)$ with parameter $\theta_U$. Its input contains two parts: parameter $\theta_D$ and the gradient of loss function with respect to $\theta_D$:

$$g_D(\theta_D, \theta_\pi) = \mathbb{E}_{s,a\sim\pi(s;\theta_\pi)}[\nabla_{\theta_D}\log\left(D(s,a;\theta_D)\right)] + \mathbb{E}_{\widetilde{s},\widetilde{a}\sim\pi_E}[\nabla_{\theta_D}\log(1 - D(\widetilde{s},\widetilde{a};\theta_D))].$$

In practice, we use a minibatch (several sample trajectories) to estimate $g_D(\theta_D, \theta_\pi)$, denoted as $\widehat{g}_D(\theta_D, \theta_\pi)$. Specifically, at the $t$-th iteration, we first calculate $\widehat{g}_D^t = \widehat{g}_D(\theta_D^t, \theta_\pi^t)$ and then update $\theta_D^{t+1} = U(\theta_D^t, \widehat{g}_D^t; \theta_U^t)$. Next, we update $\theta_U$ by gradient ascend based on the sample estimate of

$$\mathbb{E}_{s,a\sim\pi(s;\theta_\pi^t)}[\nabla_{\theta_U}\log\left(D(s,a;U(\theta_D^t,\widehat{g}_D^t;\theta_U^t))\right)] + \mathbb{E}_{\widetilde{s},\widetilde{a}\sim\pi_E}[\nabla_{\theta_U}\log(1 - D(\widetilde{s},\widetilde{a};U(\theta_D^t,\widehat{g}_D^t;\theta_U^t)))].$$

The detailed algorithm is presented in Appendix C.

**Experiments.** We compare the performance of the original GAIL and GAIL with L2L on two simulated environments: CartPole and Mountain Car (Brockman et al., 2016). As can be seen in

Figure 6, we find that GAIL has a sudden performance drop after training for a long time. We conjecture that this is because the discriminator overfits the expert trajectories and converges to a bad optimum, which is not generalizable. On the other hand, GAIL with L2L is much more stable. It is very important to real applications of GAIL: since the reward in real-world environment is usually unaccessible, we cannot know whether there is a sudden performance drop or not. With L2L, we can stabilize the training and obtain a much more reliable algorithm for real-world applications.

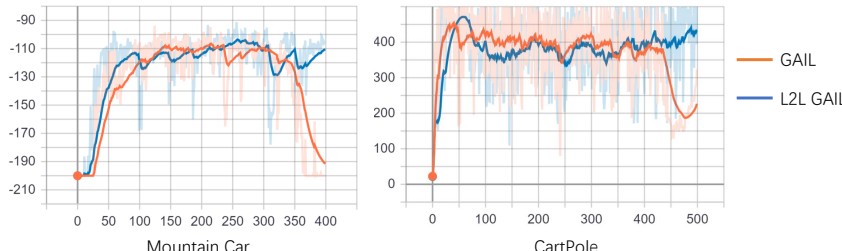

Figure 6: *Reward vs. iteration of the trained policy using original GAIL and L2L GAIL.*

## 5 DISCUSSIONS

**Closely related works**:

• By leveraging the Fenchel duality and feature embedding technique, Dai et al. (2016) convert a learning conditional distribution problem to a minimax problem , which is similar to naive attacker network. Both approaches, however, lack the primal information. In contrast, gradient attacker network considers the gradient information of primal variables, and achieves good results.

• Goodfellow et al. (2014a) propose the GAN, which is very similar to our L2L framework. Both GAN and L2L contain one generator network and one classifier network, and jointly train these two networks. There are two major difference between GAN and our framework: (1) GAN aims to transform the random noises to the synthetic data which is similar to the training examples, while ours targets on transforming the training examples to the adversarial examples for robustifying the classifier; (2) Our generator network does not only take the training examples (analogous to the random noise in GAN) as the input, but also exploits the gradient information of the objective function, since it essentially represents an optimization algorithm. The training procedure of these two, however, are quite similar. We adopt some tricks from GAN training to our framework to stabilize training process, e.g., in Grad L2L, we adopt the two-time scale trick (Heusel et al., 2017).

• There are some other works simply combining the GAN framework and adversarial training together. For example, Baluja and Fischer (2017) and Xiao et al. (2018) propose some ad hoc GAN-based methods to robustify neural networks. Specifically, for generating adversarial examples, they only take training examples as the input of the generator, which lacks the information of the outer mimnimization problem. Instead, our proposed L2L methods (e.g., Grad L2L, 2-step L2L) connect outer and inner problems by delivering the gradient information of the objective function to the generator. This is a very important reason for our performance gain on the benchmark datasets. As a result, the aforementioned GAN-based methods are only robust to simple attacks, e.g., FGSM, on simple data sets, e.g., MNIST, but fail for strong attacks, e.g., PGM and CW, on complicated data sets, e.g. CIFAR, where our L2L methods achieve significantly better performance.

**Training Stability**: For improving the training stability, we use both clean image and the corresponding gradient as the input of the attacker. Without such gradient information, the attacker severely suffers from training instability, e.g., the Naive Attacker Network. Furthermore, we try another architecture with downsampling modules, called "slim attacker" in Appendix B. We observed that the slim attacker also suffers from training instability. We suspect that the downsampling causes the loss of information. Thus, we tried to enhance the slim attacker by skip layer connections. Although the robust performance is still worse than the proposed architecture, it improves a lot.

**Benefits of our neural network approach in adversarial training**: **(1)** Since the neural network has been known to be powerful in function approximation, Our attacker network $g$ can yield strong adversarial perturbations. Since they are generated by the same attacker, the attacker $g$ essentially learns some common structures across all samples; **(2)** The attacker networks in our experiments are actually *overparametrized*, which is conjectured to ease the training of deep neural networks. We believe that similar phenomena happen to our attacker network, and ease the adversarial training.

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

# Appendix

## A BLACK-BOX ATTACK

Under the black-box setting, we first train a surrogate model with the same architecture of the target model but a different random seed, and then attackers generate adversarial examples to attack the target model by querying gradients from the surrogate model.

The black-box attack highly relies on the transferability, which is the property that the adversarial examples of one model are likely to fool others. However, the transferred attack is very unstable, and often has a large variation in its effectiveness. Therefore, results of the black-box setting might not be reliable and effective. Thus we only present one result here to demonstrate the robustness of different models.

Table 4: *Results of the black-box setting over CIFAR-10. We evaluate L2L methods with slim attacker networks.*

| Surrogate | Plain Net | | FGSM Net | | PGM Net | |
|---|---|---|---|---|---|---|
| | FGSM | PGM10 | FGSM | PGM10 | FGSM | PGM10 |
| Plain Net | 40.03 | 5.60 | 74.42 | 75.25 | 67.37 | 65.92 |
| FGSM Net | 79.20 | 85.02 | **89.90** | 80.40 | 64.28 | 63.89 |
| PGM Net | 83.80 | 84.73 | 84.33 | 85.29 | 67.05 | 65.54 |
| Naive L2L | 45.52 | 25.95 | 83.99 | 77.94 | 68.14 | 67.13 |
| Grad L2L | **86.10** | 86.87 | 87.93 | **88.01** | **71.15** | **69.95** |
| 2-Step L2L | 85.83 | **87.10** | 86.51 | 87.60 | 70.58 | 69.38 |

Table 5: Experiments under the black-box setting over CIFAR-100. Note that here we only evaluate L2L methods using the slim attacker network.

| Surrogate | Plain Net | | FGSM Net | | PGM Net | |
|---|---|---|---|---|---|---|
| | FGSM | PGM10 | FGSM | PGM10 | FGSM | PGM10 |
| Plain Net | 21.04 | 9.04 | 50.57 | 54.06 | 40.06 | 41.30 |
| FGSM Net | 42.87 | 50.73 | **61.68** | 44.70 | 39.34 | 40.08 |
| PGM Net | 56.63 | 58.34 | 56.99 | 57.97 | 40.19 | 39.87 |
| Naive L2L | 20.97 | 10.47 | 50.36 | 54.07 | 38.63 | 39.91 |
| Grad L2L | 57.63 | 59.62 | 59.18 | **61.26** | 41.71 | 41.15 |
| 2-Step L2L | **58.66** | **59.31** | 58.92 | 59.46 | **45.80** | **45.31** |

Table 6: Experiments under the black-box setting on SVHN. Note that here we only evaluate L2L methods using the wide attacker network.

| Surrogate | Plain Net | | FGSM Net | | PGM Net | |
|---|---|---|---|---|---|---|
| | FGSM | PGM10 | FGSM | PGM10 | FGSM | PGM10 |
| Plain Net | 21.72 | 6.94 | 41.81 | 33.13 | 56.77 | 49.41 |
| FGSM Net | 57.36 | 51.54 | 56.25 | 38.11 | 55.99 | 48.96 |
| PGM Net | **81.04** | **81.52** | **78.66** | 80.42 | 54.85 | 49.21 |
| Naive L2L | 73.02 | 42.14 | 78.11 | 59.79 | **85.31** | **61.08** |
| Grad L2L | 71.74 | 74.31 | 77.19 | 80.70 | 71.99 | 58.71 |
| 2-Step L2L | 65.78 | 74.07 | 76.13 | **82.80** | 61.69 | 54.13 |

## B    SLIM NETWORK

Table 7 presents another architecture that we used in the L2L. In this network, the second convolutional layer uses downsampling, while the second last deconvolutional layer uses upsampling. Due to the downsampling, this network is computationally cheap and thus it is computationally fast. For example the running time of per epoch for L2L with slim attacker is 480; whereas L2L with the original architecture is 620. However, it loses some information of input and is less stable than the original architecture (Table 1). Inspired by residual learning in He et al. (2016), we address the above issues by using a skip layer connection to ease the training of this network. Specifically, the last layer takes the concatenation of $\mathcal{A}(\boldsymbol{x}, y, \boldsymbol{\theta})$ and the output of the second last layer as input. Figure 7 presents the architecture of ResBlocks. PReLU is a special type of Leaky ReLU with a learnable slope parameter.

Table 7: *Attacker Network Architecture.*

| | |
|---|---|
| Conv: | $[k = 3 \times 3, c = 128, s = 1, p = 1]$, BN+ReLU |
| ResBlocks: | [channel = 256] |
| ResBlocks: | [channel = 128],  BN |
| DeConv: | $[k = 4 \times 4, c = 16, s = 2, p = 1]$, BN+ReLU |
| Conv: | $[k = 3 \times 3, c = 3, s = 1, p = 1]$, tanh |

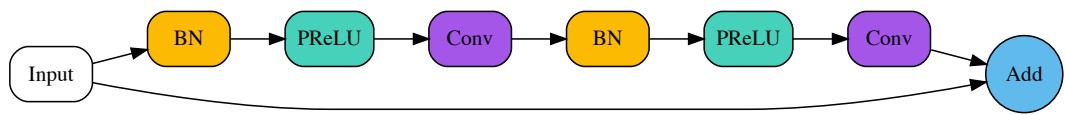

Figure 7: *An illustration example for the architecture of ResBlocks.*

Table 8 shows the results of L2L with the architecture shown in Table 7.

Table 8: *Results of L2L with the other attacker network under white-box setting over CIFAR.*

| | CIFAR-10 | | | | CIFAR-100 | | | |
|---|---|---|---|---|---|---|---|---|
| | Clean | FGSM | PGM20 | CW | Clean | FGSM | PGM20 | CW |
| Naive L2L | 94.41 | 28.44 | 0.01 | 0.00 | 75.27 | 8.47 | 0.05 | 0.00 |
| Grad L2L | 85.31 | 57.44 | 53.02 | 42.72 | 60.60 | 26.58 | 27.37 | 23.14 |
| 2-Step L2L | 75.36 | 60.19 | 46.12 | 40.82 | 60.23 | 25.92 | 20.23 | 22.70 |

## C    GENERATIVE ADVERSARIAL IMITATION LEARNING

### C.1    ALGORITHMS

---

**Algorithm 3** *Learning-to-learn-based generative adversarial imitation learning*

---

**Input:**   Expert trajectories $(\widetilde{s}, \widetilde{a} \sim \pi_E(\widetilde{s}))$, initial policy, discriminator and updater parameters $\theta_\pi, \theta_D, \theta_U$.

**for** $t \leftarrow 1$ *to* $N$ **do**

    Sample trajectories $(s, a \sim \pi(a; \theta_\pi))$

    Compute gradient:

    $g_D^t \leftarrow \frac{1}{|(s,a)|} \sum_{(s,a)} [\nabla_{\theta_D} \log (D(s, a; \theta_D^t))] + \frac{1}{|(\widetilde{s},\widetilde{a})|} \sum_{(\widetilde{s},\widetilde{a})} [\nabla_{\theta_D} \log(1 - D(\widetilde{s}, \widetilde{a}; \theta_D^t))]$

    Update the discriminator parameters:

    $\theta_D^{t+1} = U(\theta_D^t, g_D^t; \theta_U^t)$.

    Update the updater parameters $\theta_U$ by minimizing:

    $\frac{1}{|(s,a)|} \sum_{(s,a)} [\log (D(s, a; U(\theta_D^t, g_D^t; \theta_U^t)))] + \frac{1}{|(\widetilde{s},\widetilde{a})|} \sum_{(\widetilde{s},\widetilde{a})} [\log(1 - D(\widetilde{s}, \widetilde{a}; U(\theta_D^t, g_D^t; \theta_U^t)))]$.

    Update the policy parameters $\theta_\pi$ by a policy step using the TRPO rule (Ho and Ermon (2016))

---

## C.2 EXPERIMENT SETTING

**Updater Architecture** We use simple 3-layer perceptron with a skip layer as our updater. The number hidden units are $(2m \to 8m \to 4m \to m)$, where $m$ is the dimension of $\theta_D$ that depends on the original task. For the first and the second layer, we use PReLu as the activation function, while the last layer has no activation function. Finally we add the output to $\theta_D$ in the original input as the updated parameter for the discriminator network.

**Hyperparameter Settings** For all baselines we exactly follows the setting in Ho and Ermon (2016), except that we use a 2-layer discriminator with number of hidden units $((s, a) \to 64 \to 32 \to 1)$ using tanh as the activation. We use the same neural network architecture for $\pi$ and the same optimizer configuration. The expert trajectories are obtained by an expert trained using TRPO. For L2L based GAIL, we also use Adam optimizer to update the $\theta_U$ with the same configuration as updating $\theta_D$ in original GAIL.

## D LIMITING CYCLE

Limiting cycle is a well-known issue for minimax machine learning problem [4,5]. The reason behind limiting cycle is that different from minimization problems, a minimax optimization problem is more complicated and can be highly nonconvex-nonconcave, where the inner problem can not be solved exactly. Here we provide a concrete example: we consider the following optimization problem:

$$\min_x \max_y f(x, y) = xy.$$

Then at the $t$-th iteration, the update direction will be $(-y_t, x_t)$. If we start from $(1, 0)$ with a stepsize of $0.0001$, this update will result in a limiting circle: $x^2 + y^2 = 1$ and never reach the stable equilibrium $(0, 0)$ as shown in Figure 8.

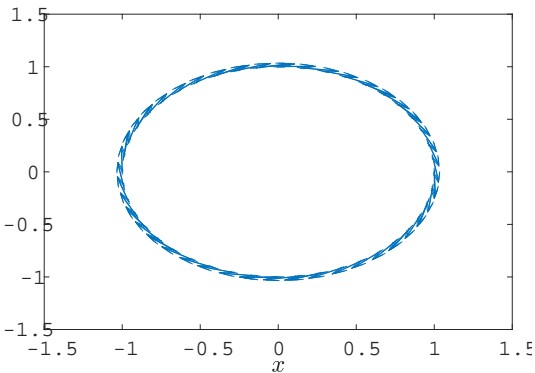

Figure 8: An example of the limiting circle: arrows denote the update directions

## E ROBUSTNESS EVALUATION CHECKLIST

Recently, there are many works on robustness defense that have been proven ineffective (Athalye et al., 2018; Carlini et al., 2019). Our work follows the most reliable and widely used robust model

approach — adversarial training, which finds a set parameters to make the model robust. We do not make any modification to final classifier model. Unlike previous works (e.g. Defense-GAN, Samangouei et al. (2018)), our model does not take the attacker as a part of the final model and does not use shattered/obfuscated/masked gradient as a defense mechanism. We also demonstrate that the evaluation of the robustness of our proposed L2L method is trustworthy by verifying all items listed in (Carlini et al., 2019).

### E.1 SHATTERED/OBFUSCATED/MASKED GRADIENT

In this section we verify that our proposed L2L method does not fall into the pitfall of shattered/obfuscated/masked gradient, which have proven ineffective. To see this, we checked every item recommended in section 3.1 of Athalye et al. (2018):

- One-step attacks perform better than iterative attacks: Figure 4 shows that the PGM attack is stronger with larger number of iterations.

- Black-box attacks are better than white-box attacks: Appendix A shows that the black-box transfer attack is much weaker than white white-box attacks.

- Unbounded attacks do not reach $100\%$ success: We evaluate the model robustness against attack with extremely large perturbation to show that unbounded attacks do reach $100\%$ success. Specifically, we use the PGM-10 attack with various perturbation magnitudes $\epsilon \in [0, 1]$ and stepsize $\frac{\epsilon}{10}$. Figure 9 shows that the PGM attack eventually reach $100\%$ success as the perturbation magnitude increases.

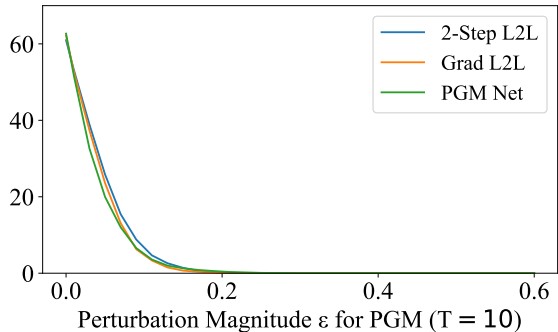

Figure 9: *Robust accuracy against perturbation magnitudes of PGM over CIFAR-100.*

- Random sampling finds adversarial examples: In Table 2, we show that random search is not better than gradient-based method and is rather weak against our model.

- Increasing distortion bound does not increase success: Figure 4 shows that the PGM attack becomes stronger as the perturbation magnitude increases.

### E.2 ROBUSTNESS EVALUATION CHECKLIST

Carlini et al. (2019) also provide an evaluation checklist, and we now check each of common severe flaws and common pitfalls as follows:

- State a precise threat model: We do not have any adversary detector; We do not use shattered/Obfuscated/Masked gradient. We do not have a denoiser. Our model has no aware of the attack mechanism, including PGM and CW attacks.

- Adaptive attacks: We used CW, PGM, and L2L attacker attack.

- Report clean model accuracy: We reported.

- Do not use Fast Gradient Sign Method. We use PGM-20 and PGM-100 and CW.

- Do not only use attacks during testing that were used during training. We use different evaluation criteria to evaluate all models.

- Perform basic sanity tests: It is provided in Figure 4.

- Generate an attack success rate vs. perturbation budget: Figure 4.

- Verify adaptive attacks perform better than any other (e.g., blackbox, and brute-force search): The above table and Appendix A in the paper.

- Describe the attacks applied: In Section 3.

- Apply a diverse set of attacks: We tried PGM attack (with different perturbation magnitude and iterations), blackbox attack (transfer attack), CW attack (adaptive attack), L2L attack (adaptive and designed for this particular model), Bruteforce random search (gradient-free attack)

- Suggestions for randomized defenses: We are not.

- Suggestions for non-differentiable components (e.g., by performing quantization or adding extra randomness): We have no additional non-differentiable component.

- Verify that the attacks have converged: Figure 4 shows that the PGM attack eventually converges.

- Carefully investigate attack hyperparameters: Figure 4

- Compare against prior work: We compared our algorithm to PDM net. L2L is more computationally efficient and the L2L model is more robust due to the fact that L2L attack is strong enough. Unlike Defense-GAN, we do not use the generator (attacker in L2L) as the denoising module and do not change the final prediction model.

