# OpenReview forum: "Learning to Defense by Learning to Attack"
_ICLR.cc/2020/Conference — Reject_

### Official Review · AnonReviewer2 · 2019-10-20
**Official Blind Review #2**

**Rating:** 3

**Review:**

In general, this paper follows the min-max training framework for adversarial robustness. Instead of using a gradient-based attack to solve the inner maximization, the authors use a neural network to learn the attack results. From the experimental results, this method can effectively defend against CW and PGD on CIFAR-10 and CIFAR-100. But the clean accuracy is lower than Madry et al.

Also, there are too many works on robustness defense that have been proven ineffective (consider the works by Carlini). Since this is a new way of robust training and there is no certified guarantee, I suggest the authors refer [1] to evaluate the effectiveness of the defense more thoroughly to convince the readers that it really works. Especially, a robustness evaluation under adaptive attacks is necessary. In other words, if the attacker knows the strategy used by the defender (such as the attacker network structure), it may be possible to break the model.

I am not convinced by the limiting cycle claim in Figure 1. I do not think this scenario (gradient descent goes along a cycle) is possible. If we take the integral of the gradient along this cycle from x to itself, we will get 0=f(x)-f(x)=-\int_{t on cycle}f'(t)dt<0, which means that the function is not continuous at x. I suggest the authors have a surface plot of the function if they think this is possible.

[1] Carlini, Nicholas, et al. "On evaluating adversarial robustness." arXiv preprint arXiv:1902.06705 (2019).

**Experience Assessment:**

I have published one or two papers in this area.

**Review Assessment: Checking Correctness Of Derivations And Theory:**

I carefully checked the derivations and theory.

**Review Assessment: Checking Correctness Of Experiments:**

I assessed the sensibility of the experiments.

**Review Assessment: Thoroughness In Paper Reading:**

I read the paper thoroughly.

---

> ### Author Response · Authors · 2019-11-14
> **To Reviewer #2 (3/3)**
>
> (2) The limiting cycle is a long-lasting issue for the minimax problem [5,6]. There might be some misunderstanding here: The reason behind the limiting cycle is that the update rule is quite different from minimization.
>
> To see the difference between the minimization problem and minimax problem, we consider the objective function $f(x,y)$ involves two variables: $x$ and $y$.
>
> | Problem         | Objective                   | Update Direction                                       | Line Integration over a cycle |
> +-------------------+----------------------------+------------------------------------------------------+-----------------------------------------+
> | Minimization | $\min_{x,y} f(x,y)$         | $(-\frac{\partial f(x,y)}{\partial x},-\frac{\partial f(x,y)}{\partial y}) = -\nabla f(x,y)$    | $\oint_{cycle} -\nabla f(x,y) d\mathbf{r} \neq 0$                |
> | Minimax        | $\min_x \max_y f(x,y)$ | $(-\frac{\partial f(x,y)}{\partial x},\frac{\partial f(x,y)}{\partial y})=   \mathbf{g}(x,y)$               | $\oint_{cycle} \mathbf{g}(x,y) \cdot d\mathbf{r} = 0$   (possible)|
>
>
>
> Figure 1 is just an illustrative example. We provide a concrete example in Appendix D in the revised version. We consider the following optimization problem: $\min_x \max_y xy$. Then at the $t$-th iteration, the update direction will be $(-y_t,x_t)$. If we start from $(1,0)$ with a small enough stepsize (e.g., $0.0001$), this update will result in a limiting circle: $x^2+y^2=1$ and never reach the stable equilibrium $(0,0)$.
>
> A general min-max optimization problem is more complicated and can be highly nonconvex-nonconcave.
>
> We have also revised the corresponding contents in the Introduction to clarify our motivation.
>
> [1] Madry, Aleksander, et al. "Towards deep learning models resistant to adversarial attacks." arXiv preprint arXiv:1706.06083 (2017).
> [2] Samangouei, P., et al. "Defensegan: Protecting classifiers against adversarial attacks using generative models." International Conference on Learning Representations, 2018
> [3] Athalye, Anish, et al. "Obfuscated Gradients Give a False Sense of Security: Circumventing Defenses to Adversarial Examples." International Conference on Machine Learning. 2018.
> [4] Carlini, Nicholas, et al. "On evaluating adversarial robustness." arXiv preprint arXiv:1902.06705 (2019).
> [5] Nagarajan, Vaishnavh, and J. Zico Kolter. "Gradient descent GAN optimization is locally stable." Advances in Neural Information Processing Systems. 2017.
> [6] Lin, Tianyi, Chi Jin, and Michael I. Jordan. "On Gradient Descent Ascent for Nonconvex-Concave Minimax Problems." arXiv preprint arXiv:1906.00331 (2019).

---

> ### Author Response · Authors · 2019-11-14
> **To Reviewer #2  (2/3)**
>
> (1.b) [4] also provide an evaluation checklist, and we respond to each of common severe flaws and common pitfalls (though many of them are basically saying the same thing) as follows:
> -- State a precise threat model: We do not have any adversary detector; We do not use shattered/Obfuscated/Masked gradient. We do not have a denoiser. Our model has no awareness of the attack mechanism, including PGM and CW attacks.
> -- Adaptive attacks: We used CW, PGM, and L2L attacker attack.
> -- Report clean model accuracy: We reported.
> -- Do not use Fast Gradient Sign Method: We are not using FGSM. We use PGM-20 and PGM-100 and CW.
> -- Do not only use attacks during testing that were used during training. We use different evaluation criteria to evaluate all models.
> -- Perform basic sanity tests: It is provided in Figure 4.
> -- Generate an attack success rate vs. perturbation budget: Figure 4.
> -- Verify adaptive attacks perform better than any other (e.g., blackbox, and brute-force search):  The above table and Appendix A in the paper.
> -- Describe the attacks applied: In Section 3 Robust Evaluation.
> -- Apply a diverse set of attacks: We tried PGM attack (with different perturbation magnitude and iterations), blackbox attack (transfer attack), CW attack (adaptive attack), L2L attack (adaptive and designed for this particular model), and Bruteforce random search.
> -- Suggestions for randomized defenses: We are not.
> -- Suggestions for non-differentiable components (e.g., by performing quantization or adding extra randomness): We have no additional non-differentiable component.
> -- Verify that the attacks have converged: Figure 4 shows that the PGM attack eventually converges.
> -- Carefully investigate attack hyperparameters: Figure 4.
> -- Compare against prior work: We compared our algorithm to PGM net. L2L is more computationally efficient and the L2L model is more robust due to the fact that L2L attack is strong enough. Unlike Defense-GAN, we do not use the generator (attacker in L2L) as the denoising module and do not change the final prediction model.
> For the special case pitfalls, including provable robust lowerbound and evaluation on other domains, are beyond the scope of this paper.

---

> ### Author Response · Authors · 2019-11-14
> **To Reviewer #2 (1/3)**
>
> (1.a) We evaluate the robustness of the network with PGM-20, PGM-100 and CW attacks, which are the state-of-the-art attack methods.
>
> We appreciate your rigorous research spirit. Indeed, evaluating the robustness of the model needs careful experimental design. The evaluation is not designed to favor our framework, but to have a fair comparison with the results in [1]. We now make a clear statement and provide more evidence to support our conclusion, which has been added to Appendix E in the revised paper.
>
> Before we make the detailed comments, we would like to make a particular response to “In other words, if the attacker knows the strategy used by the defender (such as the attacker network structure), it may be possible to break the model.” Our work follows the most reliable and widely used robust model approach — adversarial training, which finds a set of parameters to make the model robust. We do not make any modifications to the final classifier model. Unlike previous works (e.g., Defense-GAN [2]), our model does not take the attacker as a part of the final model and does not use shattered/obfuscated/masked gradient as a defense mechanism. To see this, we check each item recommended in section 3.1 of [3]:
> -- One-step attacks perform better than iterative attacks:  Figure 4 shows that the PGM attack is stronger with larger number of iterations.
> -- Black-box attacks are better than white-box attacks: Appendix A shows that the black-box attack is much weaker than white white-box attacks.  We do not make any modifications to the final classifier model. There is no reason that the black-box attack can be stronger than the white-box attack.
> -- Unbounded attacks do not reach 100% success: Figure 9 in Appendix E shows that the PGM attack eventually reaches 100% success as the perturbation magnitude increases.
> -- Random sampling finds adversarial examples: In the table below, we show that random search is not better than gradient-based methods and is rather weak against our model.
> -- Increasing distortion bound does not increase success: Figure 4 shows that the PGM attack becomes stronger as the perturbation magnitude increases.
>
> Even the attacker network structure is known, it is not obvious how to design a more effective attack.  One naive way is directly using the trained attacker model to attack the final classification model, we show that the attack is just as good as PGM attack and we have added these results in Table 2 on Page 6 in revision.
>
> CIFAR10
> | Attack                 		                    | Grad L2L    | 2-Step L2L |
> +--------------------------------------------+----------------+----------------+
> |CW                      		                    | 53.5%         | 57.07%       |
> |PGM 20               		                    | 51.17%       | 54.32%       |
> |PGM 100             		                    | 47.72%       | 52.34%       |
> |Grad L2L Attacker    		            | 49.68%       | --                 |
> |2-Step L2L Attacker		            | --       	   | 52.71%       |
> |Random (w/ $10^5$ Samples)          | 82.67%       | 83.10%       |

---

### Official Review · AnonReviewer3 · 2019-10-23
**Official Blind Review #3**

**Rating:** 6

**Review:**

The paper proposes a new way of adversarial training by placing another neural network called "attacker" network, and let the attacker to learn how to generate adversarial examples during training. This training scheme is formulated to solving a joint training according to min-max problem. Experimental results show that the method outperforms existing adversarial training in CIFAR-10/100 once the gradient information can be provided into the attacker network.

In overall, the paper is well-written with an interesting message: There are certain features useful across all data samples in the inner maximization problem, which can be induced from gradient information. The experimental results are presented clearly, demonstrating its effectiveness and efficiency in running time. Section 4 seems to support the main claim in a novel way as well. The general motivation or justification on the proposed methods, e.g. the "limiting cycle" argument or the visualization part were not that convincing or seems slightly over-claimed to me, nevertheless.

- I think adding a discussion on how to generalize the framework into other threat models, e.g. L2 or (even) unrestricted attacks would further strengthen the paper. I feel the current framework may suffers some training difficulties on these other threat models, even such kinds of discussion would also valuable to understand the method.
- As the current formulation can generalize the general inner maximization optimization process, comparing or applying the method with a more recent form of adversarial training, e.g. TRADES, would be nice to demonstrate the general applicability of the method.

**Experience Assessment:**

I have read many papers in this area.

**Review Assessment: Checking Correctness Of Derivations And Theory:**

N/A

**Review Assessment: Checking Correctness Of Experiments:**

I assessed the sensibility of the experiments.

**Review Assessment: Thoroughness In Paper Reading:**

I read the paper at least twice and used my best judgement in assessing the paper.

---

> ### Author Response · Authors · 2019-11-14
> **To Reviewer #3**
>
> (1) Thank you for your valuable comments.
> Our framework can be naturally extended to other types of attack, e.g., $\ell_2$ attacks. Figure 4 and Figure 9 in Appendix E evaluate the robustness under an unrestrictive attack, which is essentially the attack with unbounded perturbation magnitude $\epsilon=\infty$.
>
> In this paper, we focus on $\ell_\infty$ attack because it is the most reasonable evaluation and has been shown to be much more powerful than the other threat models. For example, in [2] (Figure 4), they show that to successfully attack the model,  $\ell_2$ attacks require an extremely large perturbation magnitude, which even makes the adversarial images visually **distinguishable** (The experiments only make sense when the adversarial images are visually **indistinguishable**).
>
> In terms of training stability, our framework is easy to train, since we take clean images and the corresponding gradient as the input of the attacker, which has been highlighted in our paper as a Gradient Attacker Network on Page 4. Without such gradient information, we show that such an attacker severely suffers from training instability (see Naive-L2L). We add a discussion in Section 5 (Discussion) in the revised version. Footnote 2 and Appendix B also discuss how architecture affects training stability.
>
> (2) That’s a very good suggestion.
> Our framework can be naturally extended to other types of adversarial training, including TRADES[3], VAT [4], Local Linearity Regularization [5]. We leave them for future investigation.
>
> (3) The limiting cycle is a long-lasting issue for minimax problems [6,7]. The reason behind the limiting cycle is that the update rule is quite different.
>
> To see the difference between the minimization problem and minimax problem, we consider the objective function $f(x,y)$ involves two variables: $x$ and $y$.
>
> | Problem         | Objective                   | Update Direction        |
> +-------------------+----------------------------+--------------------------------+
> | Minimization | $\min_{x,y} f(x,y)$         | $(-\frac{\partial f(x,y)}{\partial x},-\frac{\partial f(x,y)}{\partial y})$ |
> | Minimax        | $\min_x \max_y f(x,y)$ | $(-\frac{\partial f(x,y)}{\partial x},\frac{\partial f(x,y)}{\partial y})$     |
>
> Figure 1 is an illustrative example. We provide a concrete example in Appendix D in the revised version: we consider the following optimization problem: $\min_x \max_y xy$. Then at the $t$-th iteration, the update direction will be $(-y_t,x_t)$. If we start from $(1,0)$ with a small enough stepsize (e.g., $0.0001$), this update will result in a limiting circle: $x^2+y^2=1$ and never reach the stable equilibrium $(0,0)$.
>
> A general min-max optimization problem is more complicated and can be highly nonconvex-nonconcave.
>
> We have also revised the corresponding contents in the Introduction to clarify our motivation.
>
> [1]Nagarajan, Vaishnavh, and J. Zico Kolter. "Gradient descent GAN optimization is locally stable." Advances in Neural Information Processing Systems. 2017.
> [2] Poursaeed, Omid, et al. "Generative adversarial perturbations." Proceedings of the IEEE Conference on Computer Vision and Pattern Recognition. 2018.
> [3] Zhang, Hongyang, et al. "Theoretically principled trade-off between robustness and accuracy." arXiv preprint arXiv:1901.08573 (2019).
> [4] Miyato, Takeru, et al. "Virtual adversarial training: a regularization method for supervised and semi-supervised learning." IEEE transactions on pattern analysis and machine intelligence 41.8 (2018): 1979-1993.
> [5] Qin, Chongli, et al. "Adversarial Robustness through Local Linearization." arXiv preprint arXiv:1907.02610 (2019).
> [6] Nagarajan, Vaishnavh, and J. Zico Kolter. "Gradient descent GAN optimization is locally stable." Advances in Neural Information Processing Systems. 2017.
> [7] Lin, Tianyi, Chi Jin, and Michael I. Jordan. "On Gradient Descent Ascent for Nonconvex-Concave Minimax Problems." arXiv preprint arXiv:1906.00331 (2019).

---

### Official Review · AnonReviewer1 · 2019-10-25
**Official Blind Review #1**

**Rating:** 6

**Review:**

The authors propose a framework where one component is an attacker network that keeps learning about how to perturb the loss more, and one component is a defense network that robustify learning with respect to the attacker network. The framework is flexible on how the attacker network can be trained, and advances over previous works where the attacker is a human-designed algorithm rather than a learning model. Experiment results show that the framework reaches superior defense performance. The authors also extend the framework to help imitation learning.

Overall the paper is a pleasure to read. My questions/suggestions are

(1) Given that the framework seems natural in design, a deeper contribution would be talking about how to successfully train the framework in practice. The authors talk about the connection of the framework to GAN, and the latter is not that easy to train. However, we see very little information on how to train the framework in the paper. Was it super easy to train the framework (why?), or did the author encounter any difficulties? Are there important heuristics that help train the framework successfully?

(2) While the framework leads to a better defense mechanism (the authors' goal), one could wonder whether it leads to a better attacker as well. Instead of just checking the differences of the attacking examples generated, can we take the inner attacker and see if it is more effective in attacking than PGM and CW? How does the goodness of the attacker improve with time? Do different L2L variants generate attackers with different quality? Do the quality connect with the defense performance?

(3) Section 4 looks distracting to me. It is good to know that the framework can be extended to imitation learning, but the section is best put at a longer version or another paper, rather than occupying a significant amount of space in the current paper.

I have read the rebuttal and thank the authors for the response.

**Experience Assessment:**

I do not know much about this area.

**Review Assessment: Checking Correctness Of Derivations And Theory:**

I assessed the sensibility of the derivations and theory.

**Review Assessment: Checking Correctness Of Experiments:**

I assessed the sensibility of the experiments.

**Review Assessment: Thoroughness In Paper Reading:**

I read the paper at least twice and used my best judgement in assessing the paper.

---

> ### Author Response · Authors · 2019-11-14
> **To Reviewer #1**
>
> Thanks for your time and insightful suggestions.
> (1) We would like to comment on several differences between GAN training and adversarial training and several important implementation details regarding the training stability :
>
> (1.a) In GAN training, the performance heavily relies on the trade-off between the training of the discriminator and the generator, and there is no other supervision. In adversarial training, however, we have the label supervisions. Moreover, the attacker is only for generating perturbation, whereas the generator in GANs is for generating whole images from random noise. In this sense, adversarial training is easier than GAN training.
>
> (1.b) (Grad-L2L vs. Naive-L2L) For improving the training stability, we take clean images and the corresponding *gradient* as the input of the attacker, which has been highlighted in our paper as a Gradient Attacker Network on Page 4. Without such gradient information, we show that such an attacker severely suffers from training instability (see the results of Naive-L2L).
>
> (1.c) Furthermore, we want to emphasize an important detail found in our experiments (footnote 2 on page 5 and Appendix B). The network used in our framework has no downsampling modules, i.e., the image resolution does not change throughout the network. We also tried another architecture with downsampling modules, which is called “slim attacker”. We observed that the slim attacker suffers from training instability. We suspect the reason behind this is that the downsampling causes the loss of information. Thus, we tried to enhance the slim attacker by skip layer connections, by which the robust performance is still worse than the proposed architecture on CIFAR10 as illustrated in the following table.
>
> We have revised the footnote 2 and Appendix B to highlight these points, which will benefit the follow-up research.
>
> See Table 8 (Appendix B on Page 13) for full experiment results.
> | Input                  | GradL2L w/ Slim Attacker | GradL2L (Original)         |
> +----------------------+------------------------------------+---------------------------------+
> | Clean                  |   85.31%                               |  85.84%                             |
> | PGM-20 attack  |   51.02%                               |  51.17%                             |
> | CW attack          |  42.72%                                | 53.50%                              |
>
>
> Except for the aforementioned parts, we observe no other training instability issue with L2L. We have added the above discussion about the training stability to Section 5 (Discussion) on Page 8 in the revised paper.
>
> (2) As illustrated below, L2L attacker is stronger than PGM 20 and CW attacks, which explains why both Grad L2L and 2-Step L2L yield a more robust model than PGM Net.
>
> Furthermore, for Grad L2L, the perturbation generated by PGM 100 is significantly stronger than the one generated by the L2L attacker. While for 2-Step L2L, the strength of PGM 100 is similar to L2L attacker. This observation shows that 2-Step L2L attacker is stronger than Grad L2L attacker and thus yields a more robust model.
>
> | Attack                        | Grad L2L    | 2Step L2L |
> +----------------------------+----------------+--------------+
> |CW                               | 53.5%         | 57.07%     |
> |PGM 20                       | 51.17%       | 54.32%     |
> |PGM 100                     | 47.72%       | 52.34%     |
> |Grad L2L  Attacker    | 49.68%       | --               |
> |2Step L2L Attacker    | --                 | 52.71%     |
>
> The corresponding discussion has been added to Section 3 (Experiments) in the revised paper.
>
> (3) Section 4 (Extension) shows the proposed L2L framework is very generic: it can be naturally used in solving more general min-max optimization problems, such as GAIL.

---

### Decision · Program_Chairs · 2019-12-19

**Decision:**

Reject

**Comment:**

This paper considers solving the minimax formulation of adversarial training, where it proposes a new method based on a generic learning-to-learn (L2L) framework. Particularly, instead of applying the existing hand-designed algorithms for the inner problem, it learns an optimizer parametrized as a convolutional neural network. A robust classifier is learned to defense the adversarial attack generated by the learned optimizer. The idea is using L2L is sensible. However, main concerns on empirical studies remain after rebuttal.